# Synthesis, Characterization and Biological Investigation of the Platinum(IV) Tolfenamato Prodrug–Resolving Cisplatin-Resistance in Ovarian Carcinoma Cell Lines

**DOI:** 10.3390/ijms24065718

**Published:** 2023-03-16

**Authors:** Marie-Christin Barth, Norman Häfner, Ingo B. Runnebaum, Wolfgang Weigand

**Affiliations:** 1Department of Inorganic and Analytical Chemistry, Friedrich Schiller University Jena, Humboldtstrasse 8, 07743 Jena, Germany; 2Department of Gynecology and Reproduction Medicine, Jena University Hospital, Am Klinikum 1, 07747 Jena, Germany

**Keywords:** platinum(IV) prodrugs, NSAIDs, COX, cell cycle/death, resistance

## Abstract

The research on the anticancer potential of platinum(IV) complexes represents one strategy to circumvent the deficits of approved platinum(II) drugs. Regarding the role of inflammation during carcinogenesis, the effects of non-steroidal anti-inflammatory drug (NSAID) ligands on the cytotoxicity of platinum(IV) complexes is of special interest. The synthesis of cisplatin- and oxaliplatin-based platinum(IV) complexes with four different NSAID ligands is presented in this work. Nine platinum(IV) complexes were synthesized and characterized by use of nuclear magnetic resonance (NMR) spectroscopy (^1^H, ^13^C, ^195^Pt, ^19^F), high-resolution mass spectrometry, and elemental analysis. The cytotoxic activity of eight compounds was evaluated for two isogenic pairs of cisplatin-sensitive and -resistant ovarian carcinoma cell lines. Platinum(IV) fenamato complexes with a cisplatin core showed especially high in vitro cytotoxicity against the tested cell lines. The most promising complex, **7**, was further analyzed for its stability in different buffer solutions and behavior in cell cycle and cell death experiments. Compound **7** induces a strong cytostatic effect and cell line-dependent early apoptotic or late necrotic cell death processes. Gene expression analysis suggests that compound **7** acts through a stress-response pathway integrating p21, CHOP, and ATF3.

## 1. Introduction

The world-wide approvals of cisplatin and oxaliplatin by the Food and Drug Administration (FDA) smoothed the way for combating cancer with platinum(II) drugs [1,2,3,4]. Whereas cisplatin is inter alia applied in the treatment of lung, breast, and ovarian cancers [5,6,7,8,9,10], oxaliplatin is commonly used to treat pancreatic, gastric, or colorectal cancers [11,12,13,14]. Contrary to the monodentate ammine and chlorido ligands of cisplatin, oxaliplatin is equipped with chelating (1*R*,2*R*)-cyclohexane-1,2-diamine (*R*,*R*-DACH) and oxalato ligands for increased kinetic stability [1]. Both platinum(II) complexes are activated under loss of the leaving groups inside the cell by aquation. The positively charged reactive platinum(II) species further bind to the nitrogen donor atoms of the deoxyribonucleic acid (DNA) nucleobases and form crosslinks, which distort the helical structure of DNA, disrupt DNA replication and transcription, and cause apoptosis [1,15,16,17,18,19]. Despite their abundant clinical benefit, cisplatin and oxaliplatin treatment can also cause resistance and severe side-effects [15,16,19,20,21,22,23]. Platinum(IV) complexes formed by oxidative addition mainly from approved platinum(II) drugs offer the opportunity to overcome these shortcomings [4,15,16,24,25]. The octahedral coordination sphere of platinum(IV) results in higher kinetic stability compared to square-planar platinum(II) complexes and thus decreases reactions with other biomolecules apart from DNA. Inside the cell, platinum(IV) prodrugs are activated by intracellular reductants to release the cytotoxic platinum(II) complex and the free axial ligands. The axial moieties can be structurally modified influencing biological properties or can be functionalized with bioactive molecules or targeting moieties for improved pharmacological parameters and increased selectivity [1,3,4,15,16,17,24,26,27,28].

Since the development of Platin-*A* in 2014, a cisplatin-based platinum(IV) complex with aspirin as axial ligand, and the investigation of conjugates of cisplatin-based platinum(IV) complexes with ibuprofen and indomethacin, interest in the conjugation of non-steroidal anti-inflammatory drugs (NSAIDs) to the platinum(IV) center as promising prodrugs with higher efficacy and reduced side-effects increased [26,29,30,31]. Dual-action platinum(IV) complexes with one or two axial NSAID ligands and triple-action platinum(IV) complexes with another axial bioactive moiety were synthesized and tested due to their therapeutic potency [26,29,30,32,33,34,35,36,37,38,39,40,41,42,43]. NSAIDs show anti-inflammatory, antipyretic, and analgesic properties because of the inhibition of cyclooxygenases (COX) [26,44,45,46], which form eicosanoids including prostaglandins from arachidonic acid to preserve normal physiological features and are involved in the immune response [26,44,47,48,49]. One isoform of COX, COX-2, is highly expressed under inflammatory conditions and is concerned in pathological processes that cause carcinogenesis, including inhibition of apoptosis, angiogenesis, and metastasis [43,44,49,50,51,52,53]. Clinical studies to examine the potential of NSAIDs in cancer prevention showed their anti-tumor activity and preventive effects; for example, the administration of aspirin demonstrated prophylaxis in the treatment of bladder, colorectal, breast, and lung cancers [44,50,54,55,56,57]. Felbinac (FEL), part of the aryl acetic acid class of NSAIDs, is usually applied in balms for the treatment of musculoskeletal inflammatory disorders such as osteoarthritis [58,59]. Compared to oral pain-relieving NSAIDs (e.g., ibuprofen), topically used FEL displays fewer gastrointestinal side-effects but similar efficiency [58,59]. Until now, FEL has been known for its anti-inflammatory effects, but its anticancer activity has not yet been examined. Another subgroup of NSAIDs with attractive features are the fenamates, derivatives of anthranilic acid, which are orally applied painkillers. Mefenamic (MEF) and flufenamic acid (FLU) can enhance the activity of cisplatin in vitro [60,61,62]. Conjugates of cisplatin and tolfenamic acid (TOLF) assembled in nanoparticles were found to induce apoptosis and decrease metastasis in breast cancer cells in vitro and in vivo [60,63,64]. The three anthranilic acid derivatives were already examined for their anticancer activity in metal complexes including nickel(II) [65,66,67], copper(II) [66,67,68,69,70], manganese(II) [66,67], cobalt(II) [66,67,69,71], and zinc(II) [66,67,69] and showed promising results interesting for further studies.

However, the conjugation of FEL and the fenamates to a platinum(IV) complex and their effect on the cytotoxic activity has not been investigated. Thus, the synthesis and characterization of novel platinum(IV) complexes of MEF, FLU and TOLF as well as mono- and disubstituted oxaliplatin-based platinum(IV) complexes of FEL (Figure 1) were shown. The cytotoxicity of the free NSAIDs and the complexes was tested on isogenic pairs of cisplatin-sensitive and -resistant ovarian carcinoma cell lines. Cell death and cell cycle experiments were performed with the most promising platinum(IV) complex **7** containing TOLF, which was further analyzed for its stability in different buffer solutions. The ability of the (reduced) complex **7** to inhibit COX was examined in vitro. Finally, the influence of **7** on gene expression in cancer cells was examined.

The epithelial ovarian cancer (EOC) cell lines used in this study allow analysis of resistance mechanisms and are of clinical relevance because the majority of ovarian cancer patients eventually develop resistance to platinum compounds [72,73] contributing to a low survival rate below 40% [74].

**Scheme 1 ijms-24-05718-sch001:**
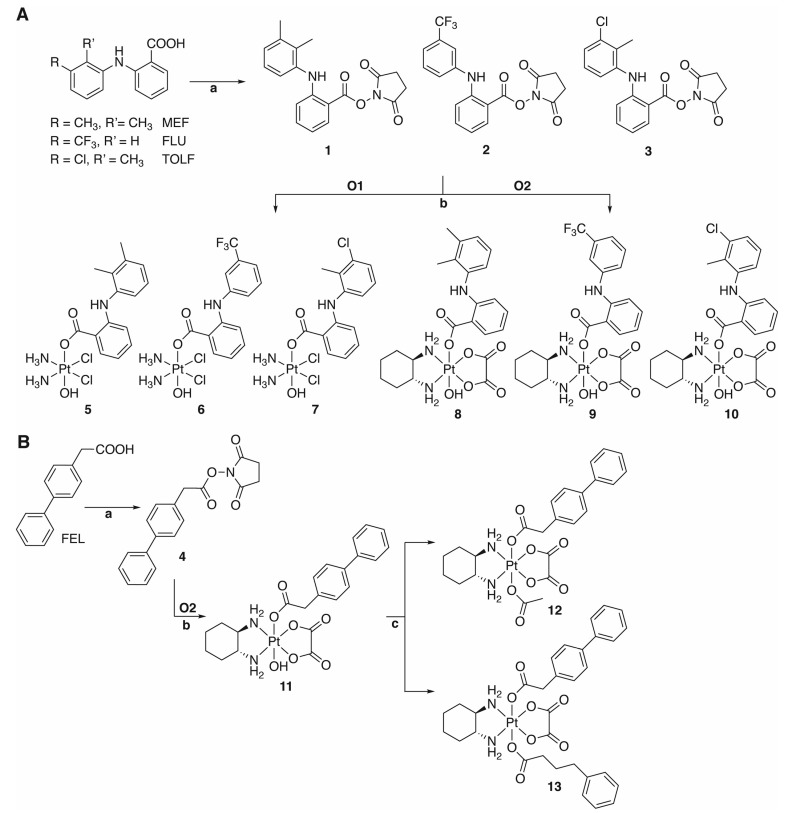
Routes for the synthesis of platinum(IV) complexes with (**A**) activated fenamate ligands (MEF: mefenamic acid, FLU: flufenamic acid, TOLF: tolfenamic acid); (**B**) activated FEL (felbinac). Reagents and conditions: (a) *N*,*N*-dicyclohexylcarbodiimide (DCC), 1-hydroxypyrrolidine-2,5-dione (NHS), chloroform, room temperature, 2–3 h; (b) dimethyl sulfoxide (DMSO), 50/65 °C, 3 d; (c) acetic anhydride or phenyl butyric anhydride, dimethyl fromamide (DMF), room temperature, 3 d [36,75,76].

## 2. Results and Discussion

### 2.1. Synthesis of the Active Esters **1**–**4**

Due to the low reactivity of carboxylic acids, the NSAIDs were activated by *N*,*N*-dicyclohexylcarbodiimide (DCC) and 1-hydroxypyrrolidine-2,5-dione (NHS) in chloroform to form active esters with succinimidyl moiety as a leaving group (Figure 1) [75]. The byproduct dicyclohexylurea is insoluble in chloroform and can be filtered off. The active esters **1**–**4** were purified by column chromatography (chloroform/cyclohexane) and were characterized by nuclear magnetic resonance (NMR) spectroscopy, mass spectrometry, and elemental analysis. The structures of the ligands are shown in Figure 1. The active ester formation was monitored by ^1^H and ^13^C{^1^H} NMR spectroscopy (Appendix A). The methylene protons of the active ester’s succinimidyl moiety appear at 2.84–2.95 ppm, apart from other signals. In the case of ligand **4**, besides the phenol protons in the aromatic region, the methylene protons at 3.99 ppm are noticeable. The amine protons of ligands **1**–**3**, as derivatives of anthranilic acid, are shifted to a lower field around 8–9 ppm. The methyl protons of ligands **1** and **3** are detected as singlets at 2.33 and 2.15 ppm (ligand **1**, Appendix A) and 2.29 ppm (ligand **3**, Appendix A).

### 2.2. Synthesis of the Platinum(IV) Complexes **5**–**13**

Cisplatin was synthesized in several steps from K_2_PtCl_4_ according to literature [77]. The oxidation of cisplatin and oxaliplatin to **O1** and **O2** was performed with hydrogen peroxide (Appendix A) [36,78]. The ensuing reactions with the active esters **1**–**4** proceeded in dimethyl sulfoxide (DMSO) at 50–65 °C to the monosubstituted platinum(IV) complexes **5**–**11** [36]. Further modification of complex **11** for the synthesis of disubstituted platinum(IV) complexes was applied by reactions with acid anhydrides in dimethyl formamide (DMF) at room temperature [76]. The acetato group as a non-bioactive ligand [15,16,17] was chosen to enhance the solubility of complex **12** compared to the hydroxido ligand, while phenyl butyric anhydride was chosen as histone deacetylase inhibitor to increase the cytotoxicity of the platinum(IV) complex (**13**) [41,79,80,81]. All compounds were purified by column chromatography (chloroform/methanol) and were characterized by NMR spectroscopy, mass spectrometry, and elemental analysis (see Section 3.2 and Appendix A). Complexes **5**–**10** with axial fenamato ligands were obtained in yields from 5 to 15%, depending on the platinum(II) core and the fenamate. Higher yields were achieved using FEL as an axial ligand (33–38% for complexes **11**–**13**). The ^195^Pt chemical shifts vary depending on the coordination sphere from around 1020 ppm for cisplatin-based platinum(IV) complexes (**5**–**7**), 1400 ppm for monosubstituted oxaliplatin-based platinum(IV) complexes (**8**–**11**), to 1600 ppm for disubstituted complexes **12** and **13**. The characteristic amine protons of complexes **5**–**10** are detected at around 9 ppm as singlet, while the methyl groups of complexes **5**, **7**, **8,** and **10** are shown at around 2 ppm. The resonance signal at 3.57 ppm is assigned to methylene protons of the platinum(IV) complex with axial FEL (**11**). The further reactions with acetic and phenyl butyric anhydride (**12** and **13**) lead to splitting of the methylene protons into doublets at 3.64 and 3.65 ppm.

### 2.3. Anticancer Activity

The cytotoxicity of the platinum(IV) complexes **5**–**8** and **10**–**13**, as well as NSAIDs, was tested on the ovarian carcinoma cell lines A2780par and SKOV3par as well as their cisplatin-resistant analogues A2780cis and SKOV3cis [82]. Complex **9** was not tested due to its nominal yield. Cells were seeded on 96-well plates and were incubated for 24 h in medium. Cisplatin and oxaliplatin were examined as referential platinum(II) complexes. Both compounds were dissolved in 0.9% NaCl solution right before the experiment. The platinum(IV) complexes were dissolved and serially diluted in DMSO, finally diluted in Roswell Park Memorial Institute (RPMI) 1640 medium to reach concentrations from 0.1 to 100 μM, and added to the cells. Despite the cytotoxic effect of DMSO on cells, it is usually used as a solvent in biological tests [83]. To prevent damage to cells, the concentration of DMSO in cell culture experiments was limited to 0.1%. Additional triplets of wells with medium and medium with 0.1% DMSO functioned as control. Half-maximal inhibitory concentration (IC_50_) values were determined by 2-(4,5-Dimethyl-1,3-thiazol-2-yl)-3,5-diphenyl-3H-1,2λ^5^,3,4-tetrazol-2-ylium bromide (MTT) assay after an exposure of 48 h from at least three independent experiments (Table 1). The metabolic activity of cells is examined by the conversion of the MTT dye into formazan and was measured by absorbance at 570 nm [84,85].

Whereas the free NSAIDs do not cause appreciable cytotoxicity (IC_50_ > 100 μM, Table 1), when attached to a platinum(II) core, the resulting platinum(IV) complexes cause much lower IC_50_ values in the cancer cell lines tested (Table 1). A2780 cell lines exhibit higher sensitivity to the platinum(IV) complexes than cells from SKOV3. Comparing the IC_50_ values of the complexes **5**–**7** against **8** and **10**, the oxaliplatin-based platinum(IV) complexes are less cytotoxic than the complexes with a cisplatin-core (Table 1). Especially, complexes **5** and **7** feature high antiproliferative activity in the selected cancer cell lines, and they are even more active than cisplatin. Remarkably, **5** and **7** have a higher effect on cisplatin-resistant cell lines (**5**: resistance factor (RF) 0.9 for A2780 and RF 0.3 for SKOV3; **7**: RF 0.5 for A2780 and RF 0.3 for SKOV3, Table 1). The mono-substituted platinum(IV) complex with axial FEL **11** shows low cytotoxicity in A2780 cell lines, whereas the IC_50_ values in SKOV3 cell lines are higher than the examined concentration range. Further substitution of **11** with acetic anhydride improves the antiproliferative activity of complex **12** in A2780 cell lines, potentially because of the increased lipophilic character of the acetate group compared to the hydroxido ligand (Table 1) [86,87,88]. Considerably decreased IC_50_ values and thus higher activity are achieved by the reaction of **11** with phenyl butyric anhydride (complex **13**, Table 1). The phenyl butyrato ligand enhances the lipophilicity and additionally functions as histone deacetylase inhibitor inducing cell death and cell cycle arrest in cancer cells [89,90]. Compound **13**, which consists of the cytotoxic oxaliplatin-core, and two different biologically active ligands, belongs to the group of platinum(IV) complexes with triple-action. Hereby, cancer cells are attacked in three different ways, which improves the cytotoxicity compared to the mono-substituted oxaliplatin(IV) complexes (Table 1).

Due to its promising antiproliferative activity in the tested cancer cell lines (also slightly higher than that of complex **5**), complex **7** was selected for further biological experiments. Besides a high cytotoxicity against tumor cells, chemotherapeutics should have a low activity against normal cells potentially resulting in reduced side-effects and the application of higher concentrations in vivo. Thus, we tested **7** on primary fibroblast cultures. Compound **7** showed a high cytotoxic activity with an IC_50_ value of 0.68 +/− 0.03 µM. The selectivity index (SI = mean IC_50_^Fibroblasts^/IC_50_^EOC^) was 1.47, whereas cisplatin showed an IC_50_ of 8.26 +/− 1.1 µM and an SI of 0.58. Compound **7** is slightly more specific for EOC compared to cisplatin. Similar SI were observed for an oxaliplatin-based platinum(IV) prodrug with naproxen [37]. However, additional experiments are necessary to evaluate the selectivity, to detect side effects, and to estimate a clinically useful concentration. These experiments may consist of 3D-co-culture or mouse models [91,92].

### 2.4. Investigations on Stability Behavior

To analyze the stability behavior of complex **7**, UV-Vis spectroscopic experiments were performed at 37 °C in phophate-buffered saline (PBS), 120 mM NaCl (extracellular concentration), 12 mM NaCl (intracellular concentration), and DMSO, as well as bovine serum albumin (BSA) protein and salmon sperm DNA over time (Appendix A). Complex **7** shows overall stability in PBS, 12 mM NaCl, BSA, and DMSO over 23 h. In 120 mM NaCl solution, the absorption decreases over time without precipitation of **7**, indicating slow degradation. Slight changes in the absorption are observed in DNA solution, where the absorption maxima around 280 nm are marginally blue shifted.

The reaction of **7** with ten-fold excess of ascorbic acid in aqueous phosphate buffer (5 mM, pH 7.4) was analyzed by ultra-high performance liquid chromatography high-resolution mass spectrometry (UHPLC-HRMS). After 5 min, over 40% of the complex is degraded (Appendix A). The further decomposition of **7** proceeds within 5 h. Simultaneously, the free ligand TOLF and cisplatin are formed progressively during the reaction, confirming the reduction of platinum(IV).

### 2.5. In Vitro Inhibition of COX Activity

The main principle of platinum(IV) prodrugs is based on the intracellular reduction of platinum(IV) resulting in the release of the biologically active axial ligands and generation of the original platinum(II) drug [1,3,15]. The influence of complex **7** and the reduced compound **7** on COX activity was analyzed in vitro. Whereas compound **7** could not inhibit COX-1 activity, the free ligand (TOLF) and the reduced complex **7** inhibited COX-1 activity significantly by 84.4% and 47.4%, respectively (Appendix A; *p* < 0.005). Thus, the ligand TOLF must be released after reduction to platinum(II) to inhibit COX. However, it cannot be excluded that the conditions of this in vitro assay introduce bias. Albeit ascorbic acid alone does not inhibit COX activity (Appendix A) and compound **7** was reduced before the addition to the COX reaction, another interaction of ascorbic acid with the assay system or compound **7** may cause the COX inhibition. Additionally, the used assay seems to be suboptimal because high concentrations (80 µM) were required to inhibit COX-1, and no influence on COX-2 activity was observed at this concentration (Appendix A). This may be caused by suboptimal buffer conditions or incubation times to reach a high inhibition of COX. Similarly, Khoury et al. recently reported a COX-2 inhibitory effect of aspirin with this commercial kit at high concentrations (700 µM) only [93]. Moreover, in this study the authors observed no strong correlation between the cytotoxicity of platinum(IV) compounds and their lipophilicity or COX-2 inhibition strength [93]. Albeit we can prove the successful COX-1 inhibition by the reduced compound **7** (Appendix A), IC_50_ data from the combination treatment cisplatin + TOLF (Table 1) do not show an increased cytotoxicity compared to the cisplatin treatment alone. Thus, the high cytotoxicity of **7** is potentially not directly related to COX inhibition by TOLF. A COX-independent high cytotoxicity was similarily observed by Ravera et al. analyzing platinum(IV) prodrugs with ketoprofen and naproxen [35]. Thus, the tumor cell-specific effects seem to be independent from COX inhibition. However, COX inhibition has anti-inflammatory effects in vivo that may influence tumor properties.

### 2.6. Cell Death and Cell Cycle Distribution Analyses

Cell cycle and cell death experiments were performed by flow-cytometry assisted by propidium iodide (PI) staining of DNA. Since PI can only enter dead cells and binds stoichiometrically to nucleic acids, the measured emission informs about the number of dead cells and is proportional to the DNA content, thus indicating the cell cycle phase of individual cells [94,95]. The percentage of cell death indicates the fraction of dead cells in the population. As assumed, the percentage of dead cells shows a concentration-dependent increase upon treatment with complex **7** and significant differences compared to untreated cells (Figure 1A, *p* < 0.05). SKOV3 cell lines are more strongly influenced by cell death induction and show between 12.1% and 34.1% dead cells. In contrast, A2780 cell lines show lower cell death between 3.8% and 6.2% albeit, treated with similarly effective concentrations as determined by the MTT assay (IC_50_, two-fold IC_50_). However, the number of PI-negative, single-cell events by flow-cytometry is vigorously decreased after treatment, indicating decreased proliferation induced by cytostatic effects of complex **7** specifically in A2780 (Figure 1B, *p* < 0.05). Differences in the cell death or number of alive cells between parental and cisplatin-resistant cells were not significant, whereas cisplatin treatment induced significantly higher cell death in parental compared to resistant cells (Appendix A, *p* < 0.01). Thus, complex **7** activity is not affected by cisplatin-resistance mechanisms.

Figure 2 shows the cell cycle phases depending on the treatment with compound **7** for each cell line. The majority of untreated cells stays in G1 phase, which decreases continuously with higher concentrations of complex **7**. The treatment with at least 0.2 µM compound **7** leads to a significant cell cycle arrest in G2/M in all cell cultures and in the S phase except for A2780par (Figure 2). Complex **7** influences the cell cycle distribution more strongly in SKOV3 compared to A2780. This may cause the observed higher cell death in SKOV3 vs. A2780 (Figure 1).

### 2.7. Investigation of Apoptosis/Necrosis

To analyze induced cell death, apoptosis, and necrosis were measured in control and compound **7**-treated cell lines using a commercial kit (see Section 3.7). This assay detects apoptotic and necrotic processes by live-cell real-time measurements of luminescence and fluorescence signals reflecting the presence of phosphatidylserine on the cell membrane surface or the accessibility of genomic DNA for a cell-impermeable dye, respectively.

The influence of complex **7** (3 µM) on cell death of A2780 and SKOV3 cells (both parental and cisplatin-resistant) was analyzed for 70 h (Figure 3). All tested cell lines show cell death by apoptosis followed by secondary necrosis, and no general differences between parental and cisplatin-resistant cells were detected. Higher levels of (early) apoptosis for SKOV3 compared to A2780 cells resemble flow-cytometry-based cell death analyses (Figure 1). A2780 cells show high levels of necrosis at later time points not analyzed by flow-cytometry. Resistant SKOV3 cells seem to respond earlier and with higher apoptosis induction compared to parental cells, validating the activity of complex **7** against cisplatin-resistant cells (RF < 1; Table 1). The exact cause for the decline of the luminescence signal after 24 h to 30 h is unknown. The commercial system uses a time-released luciferase substrate that must be converted by a cellular esterase from alive cells. Thus, a reduction of the cell number by treatment may led to a decreased conversion of the substrate.

### 2.8. Gene Expression Analyses

The expression of specific genes after drug treatment can give information about its mechanism of action [96,97]. Thus, we measured gene expression of the cell cycle inhibitor p21, of the stress response factor and apoptosis inductor CCAAT/enhancer-binding proteins homologous protein (CHOP, also known as DNA damage inducible transcript 3, DDIT3), and of the activating transcription factor 3 (ATF3) that is involved in various stress responses and a potential COX-independent target of NSAID [98] by real-time polymerase chain reaction (PCR). As shown in Figure 4, complex **7** induces a strong *p21* expression both in cisplatin-sensitive and -resistant A2780 cells, whereas cisplatin at IC_50_ of sensitive cells induced a lower *p21* expression in resistant cells. Cisplatin treatment of resistant cells with IC_50_ concentration induced *p21* to identical levels as complex **7** (Figure 4). The similar *p21* overexpression under complex **7** treatment agrees with the observed identical changes of the cell cycle distribution between sensitive and resistant cells after complex **7** treatment (Figure 2). Thus, complex **7** can induce a p21-dependent cell cycle arrest both in cisplatin-sensitive and resistant cells at concentrations ≤1 µM. In addition, complex **7** promotes an upregulation of the apoptosis inducer *CHOP* that is overexpressed by various genotoxic agents in sensitive cells [99]. The stronger upregulation in SKOV3 compared to A2780 cells resembles the higher levels of apoptosis and cell death in this cell line (Figure 2 and Figure 4). CHOP can be induced during endoplasmatic reticulum stress (ER stress) by ATF4 [100]. However, an increased splicing of *Xbp1* indicative of unfolded protein response-induced ER stress [101] was not observed under complex **7** treatment, pointing to genotoxic or direct NSAID-based activation of the ER stress pathway. COX inhibitors can induce ATF3, an ER stress pathway gene mediating apoptosis in colorectal cancer cells and ferroptosis in gastric cancer cells [102,103,104]. Thus, complex **7** may activate the ER stress pathway similarly in ovarian cancer cells. As shown in Figure 4, *ATF3* is upregulated by complex **7** both in cisplatin-sensitive and resistant A2780 and SKOV3 cells. This upregulation is increased in resistant cells compared to cisplatin treatment. Cisplatin induces less *ATF3* in resistant cells, confirming data from gastric cancer where ATF3 expression mediates cisplatin sensitivity [104]. Moreover, *ATF3* induction by compound **7** is concentration-dependent, whereas this effect is not seen for cisplatin at the analyzed concentrations (Figure 4). Altogether, these experiments point to the contribution of the ER stress pathway to the biologic activity of complex **7** and to the potential strategy to target this pathway for resolving platinum-resistance. However, additional experiments are necessary to (i) enable a statistical evaluation, which is not possible with the presently available data of n = 2 independent treatments and (ii) to determine the exact mechanism for the activation of the ER stress pathway by compound **7**.

## 3. Materials and Methods

### 3.1. Materials and Techniques

All reactions were carried out under atmospheric conditions. The chemicals and solvents used were commercially available and used without further purification. K_2_PtCl_4_ and oxaliplatin were obtained from Umicore AG & Co. KG (Hanau-Wolfgang, Germany). Chemicals were commercially available (TCI, Eschborn, Germany; abcr, Karlsruhe, Germany; Acros Organics, Niderau, Germany; Carl Roth, Karlsruhe, Germany). Solvents of technical grade were distilled prior to their use. Silica gel (0.063–0.2 mm) was used for column chromatography, and thin-layer chromatography (TLC) was carried out using TLC aluminum sheets from Merck (Silica gel 60 F_254_). ^1^H NMR, ^13^C NMR, ^195^Pt NMR, and ^19^F spectra were recorded with a Bruker Avance 200 MHz, 400 MHz, or 600 MHz NMR spectrometer (Bruker, Billerica, MA, USA). Mass spectrometric techniques are reported in detail in the Appendix A). Elemental analyses were performed with a Leco CHNS-932 device (Leco, Mönchengladbach, Germany). UV-Vis spectroscopic experiments were monitored with a JASCO UV/VIS V-760-ST spectrophotometer (JASCO, Pfungstadt, Germany). Absorption spectra were measured from 240 to 800 nm with 1 nm steps and a scan speed of 400 nm/min. The measurements were normalized to the respective buffer. Cell death and cell cycle analyses were performed with a BD Accuri^TM^ C6 Plus flow-cytometer (BD Franklin Lakes, NJ, USA). Apoptosis/necrosis was tested with the RealTime-Glo^TM^ Annexin V Apoptosis and Necrosis Assay (Promega, Walldorf, Germany). COX inhibition was examined by COX (ovine/human) Inhibitor Screening Assay by Cayman Chemical (Ann Arbor, MI, USA). Real-time PCR experiments were run on a Rotorgene cycler (Qiagen, Hilden, Germany).

### 3.2. Syntheses

Oxoplatin ([Pt(NH_3_)_2_Cl_2_(OH)_2_], **O1**) and oxidized oxaliplatin ([Pt(DACH)(OH)_2_(ox)_2_], **O2**) were synthesized following literature procedures [36,77,78].

General procedure 1 for the activation of carboxylic acids:

Carboxylic acid (1 equiv.), DCC (1.1 equiv.), and NHS (1 equiv.) were dissolved in chloroform and stirred for 2–3 h at room temperature. After filtration, the filtrate was evaporated under reduced pressure, and the raw product was purified by column chromatography (chloroform/cyclohexane).

Active ester of MEF (**1**). Synthesis was performed according to general procedure 1. Mefenamic acid (400 mg, 1.66 mmol), DCC (376 mg, 1.82 mmol), NHS (191 mg, 1.66 mmol). Yield: 300 mg (53.4%) as yellow solid. ^1^H NMR (CD_2_Cl_2_, 400 MHz): *δ* = 8.66 (s, 1 H), 8.08 (dd, *J* = 8.2/1.3 Hz, 1 H), 7.35 (m, 1 H), 7.11 (m, 3 H), 6.71 (m, 2 H), 2.89 (s, 4 H), 2.33 (s, 3 H), 2.15 (s, 3 H) ppm. ^13^C{^1^H} NMR (CD_2_Cl_2_, 101 MHz): *δ* = 170.2, 163.9, 151.6, 139.1, 138.2, 136.9, 133.8, 131.7, 128.3, 126.7, 124.5, 117.0, 114.5, 105.7, 26.3, 20.8, 14.3 ppm. MS (DEI): *m*/*z* 338 [M]^·+^. EA: calc. for C_19_H_18_N_2_O_4_·1/3 EtOAc: C: 66.41%; H: 5.67%; N: 7.62%, found: C: 66.10%; H: 5.44%; N: 8.10%.

Active ester of FLU (**2**). Synthesis was performed according to general procedure 1. Flufenamic acid (400 mg, 1.42 mmol), DCC (323 mg, 1.56 mmol), NHS (164 mg, 1.42 mmol). Yield: 215 mg (40.0%) as amber solid. ^1^H NMR (CDCl_3_, 400 MHz): *δ* = 8.95 (s, 1 H), 8.16 (m, 1 H), 7.45 (m, 5 H), 7.24 (m, 1 H), 6.87 (t, *J* = 7.6 Hz, 1 H), 2.95 (s, 4 H) ppm. ^13^C{^1^H} NMR (CDCl_3_, 101 MHz): *δ* = 169.4, 163.2, 148.3, 140.4, 136.5, 131.9, 130.1, 125.7, 120.8, 120.8, 119.3, 119.3, 118.5, 114.1, 107.2, 25.7 ppm. MS (DEI): *m*/*z* 378 [M]^·+^. EA: calc. for C_18_H_13_F_3_N_2_O_4_ ∙ 1/10 CHCl_3_: C: 55.71%; H: 3.38%; N: 7.18%, found: C: 55.96%; H: 3.47%; N: 7.10%.

Active ester of TOLF (**3**). Synthesis was performed according to general procedure 1. Tolfenamic acid (400 mg, 1.53 mmol), DCC (347 mg, 1.68 mmol), NHS (176 mg, 1.53 mmol). Yield: 280 mg (51.0%) as amber solid. ^1^H NMR (CD_2_Cl_2_, 400 MHz): *δ* = 8.70 (s, 1 H), 8.11 (m, 1 H), 7.40 (m, 1 H), 7.22 (m, 3 H), 6.79 (m, 2 H), 2.90 (s, 4 H), 2.29 (s, 3 H) ppm. ^13^C{^1^H} NMR (CD_2_Cl_2_, 101 MHz): *δ* = 170.1, 164.0, 150.6, 140.0, 137.0, 136.1, 133.1, 131.9, 127.7, 127.2, 124.8, 117.9, 114.6, 106.5, 26.3, 15.4 ppm. MS (DEI): *m*/*z* 358 [M]^·+^. EA: calc. for C_18_H_15_ClN_2_O_4_ ∙ 1/5 CHCl_3_: C: 57.34%; H: 4.08%; N: 7.27%, found: C: 57.58%; H: 4.12%; N: 7.24%.

Active ester of FEL (**4**). Synthesis was performed according to general procedure 1. Felbinac (400 mg, 1.88 mmol), DCC (428 mg, 2.07 mmol), NHS (217 mg, 1.88 mmol). Yield: 350 mg (60.2%) as white solid. ^1^H NMR (CDCl_3_, 400 MHz): *δ* = 7.60 (m, 4 H), 7.44 (m, 4 H), 7.36 (m, 1 H), 3.99 (s, 2 H), 2.84 (s, 4 H) ppm. ^13^C{^1^H} NMR (CDCl_3_, 151 MHz): *δ* = 169.0, 166.7, 140.8, 140.5, 130.3, 129.7, 128.8, 127.6, 127.4, 127.1, 37.3, 25.6 ppm. MS (DEI): *m*/*z* 309 [M]^·+^. EA: calc. for C_18_H_15_NO_4_: C: 69.89%; H: 4.89%; N: 4.53%, found: C: 69.90%; H: 5.09%; N: 4.39%.

General procedure 2 for the synthesis of platinum(IV) complexes:

**O1** or **O2** (1 equiv.) and the active ester (1.1 equiv.) were suspended in DMSO (6 mL) and stirred at 50–65 °C for 3 d. The solvent was removed by stepwise addition and removal of excess ether. The raw product was purified by column chromatography (chloroform/methanol).

*Cis,cis,trans*-[Pt(NH_3_)_2_Cl_2_(MEF-H)OH] (**5**). Synthesis was performed according to general procedure 2. **O1** (200 mg, 0.6 mmol), **1** (184 mg, 0.66 mmol). Yield: 50 mg (15.0%) as yellow solid. ^1^H NMR (DMSO-d_6_, 400 MHz): *δ* = 9.29 (s, 1 H), 7.85 (d, *J* = 6.6 Hz, 1 H), 7.17 (t, *J* = 7.1 Hz, 1 H), 7.05 (m, 2 H), 6.95 (m, 1 H), 6.64 (m, 2 H), 6.03 (m, 6 H), 2.27 (s, 3 H), 2.14 (s, 3 H) ppm. ^13^C{^1^H} NMR (DMSO-d_6_, 101 MHz): *δ* = 174.7, 146.9, 139.5, 137.5, 132.4, 132.1, 131.1, 125.6, 125.4, 121.4, 117.1, 115.8, 112.9, 20.3, 14.2. ^195^Pt{^1^H} NMR (DMSO-d_6_, 86 MHz): *δ* = 1024 ppm. ESI-HRMS^+^: *m*/*z* [M+H]^+^ calc. for C_15_H_22_Cl_2_N_3_O_3_^194^Pt: 556.0660, found: 556.0655. EA: calc. for C_15_H_21_Cl_2_N_3_O_3_Pt ∙ 1/7 CHCl_3_: C: 31.67%; H: 3.71%; N: 7.32%, found: C: 31.68%; H: 3.94%; N: 7.65%.

*Cis,cis,trans*-[Pt(NH_3_)_2_Cl_2_(FLU-H)OH] (**6**). Synthesis was performed according to general procedure 2. **O1** (100 mg, 0.3 mmol), **2** (128 mg, 0.33 mmol). Yield: 26 mg (14.6%) as yellow solid. ^1^H NMR (DMSO-d_6_, 400 MHz): *δ* = 9.81 (s, 1 H), 7.86 (d, *J* = 7.6 Hz, 1 H), 7.50 (m, 3 H), 7.37 (m, 2 H), 7.23 (d, *J* = 6.5 Hz, 1 H), 6.85 (m, 1 H), 6.12 (m, 6 H) ppm. ^13^C{^1^H} NMR (DMSO-d_6_, 101 MHz): *δ* = 173.8, 143.1, 142.8, 132.2, 132.2, 130.3, 130.2, 129.9, 125.6, 122.9, 121.8, 120.2, 118.7, 117.1, 114.9, 114.7. ^195^Pt{^1^H} NMR (DMSO-d_6_, 86 MHz): *δ* = 1032 ppm. ^19^F{^1^H} NMR (DMSO-d_6_, 376 MHz): *δ* = −61.34 ppm. ESI-HRMS^+^: *m*/*z* [M+H]^+^ calc. for C_14_H_17_Cl_2_F_3_N_3_O_3_^194^Pt: 596.0221, found: 596.0237. EA: calc. for C_14_H_16_Cl_2_F_3_N_3_O_3_Pt · CH_3_CN: C: 30.11%; H: 3.00%; N: 8.78%, found: C: 30.59%; H: 3.25%; N: 8.31%.

*Cis,cis,trans*-[Pt(NH_3_)_2_Cl_2_(TOLF-H)OH] (**7**). Synthesis was performed according to general procedure 2. **O1** (100 mg, 0.3 mmol), **3** (118 mg, 0.33 mmol). Yield: 25 mg (14.4%) as yellow solid. ^1^H NMR (DMSO-d_6_, 400 MHz): *δ* = 9.44 (s, 1 H), 7.91 (m, 1 H), 7.26 (m, 2 H), 7.17 (m, 2 H), 6.80 (m, 2 H), 6.07 (m, 6 H), 2.33 (s, 3 H) ppm. ^13^C{^1^H} NMR (DMSO-d_6_, 101 MHz): *δ* = 174.2, 145.5, 141.6, 134.3, 132.4, 132.1, 129.3, 127.1, 123.7, 120.6, 118.2, 117.1, 113.7, 15.3 ppm. ^195^Pt{^1^H} NMR (DMSO-d_6_, 86 MHz): *δ* = 1021 ppm. ESI-HRMS^−^: *m*/*z* [M−H]^−^ calc. for C_14_H_17_Cl_3_N_3_O_3_^194^Pt: 573.9968, found: 573.9962. EA: calc. for C_14_H_18_Cl_3_N_3_O_3_Pt: C: 29.10%; H: 3.14%; N: 7.27%, found: C: 28.86%; H: 3.31%; N: 7.64%.

*Cis,cis,trans*- [Pt(DACH)(ox)(MEF-H)OH] (**8**). Synthesis was performed according to general procedure 2. **O2** (150 mg, 0.35 mmol), **1** (130 mg, 0.38 mmol). Yield: 30 mg (13.1%) as yellow solid. ^1^H NMR (DMSO-d_6_, 400 MHz): *δ* = 9.30 (s, 1 H), 8.33 (m, 1 H), 8.24 (m, 1 H), 7.96 (m, 1 H), 7.83 (dd, *J* = 8.2/1.6 Hz, 1 H), 7.20 (m, 2 H), 7.08 (m, 2 H), 6.97 (m, 1 H), 6.63 (m, 2 H), 2.69 (m, 3 H), 2.27 (s, 3 H), 2.14 (m, 2 H), 2.08 (s, 3 H), 1.48 (m, 4 H), 1.16 (m, 2 H) ppm. ^13^C{^1^H} NMR (DMSO-d_6_, 101 MHz): *δ* = 176.0, 164.0, 147.3, 139.0, 137.7, 132.9, 132.5, 131.0, 125.9, 121.6, 116.1, 115.9, 113.0, 61.5, 60.3, 30.9, 30.7, 23.7, 23.6, 20.3, 13.9 ppm. ^195^Pt{^1^H} NMR (DMSO-d_6_, 86 MHz): *δ* = 1416 ppm. ESI-HRMS^+^: *m/z* [M+H]^+^ calc. for C_23_H_30_N_3_O_7_^194^Pt: 654:1705, found: 654.1703. EA: calc. for C_23_H_29_N_3_O_7_Pt · 1/4 CHCl_3_: C: 40.80%; H: 4.31%; N: 6.14%, found: C: 40.96%; H: 4.43%; N: 6.14%.

*Cis,cis,trans*-[Pt(DACH)(ox)(FLU-H)OH] (**9**). Synthesis was performed according to general procedure 2. **O2** (125 mg, 0.29 mmol), **2** (121 mg, 0.32 mmol). Yield: 10 mg (5.0%) as yellow solid. ^1^H NMR (DMSO-d_6_, 400 MHz): *δ* = 9.64 (s, 1 H), 8.19 (m, 2 H), 7.99 (s, 1 H), 7.90 (d, *J* = 7.8 Hz, 1 H), 7.50 (m, 2 H), 7.37 (t, *J* = 7.4 Hz, 1 H), 7.28 (t, *J* = 6.6 Hz, 2 H), 7.23 (s, 1 H), 6.85 (t, *J* = 7.5 Hz, 1 H), 2.77 (s, 1 H), 2.62 (m, 2 H), 2.08 (dd, J = 25.4/11.3 Hz, 2 H), 1.46 (m, 4 H), 1.09 (m, 2 H) ppm. ^13^C{^1^H} NMR (DMSO-d_6_, 101 MHz): *δ* = 174.9, 163.9, 143.9, 142.3, 132.9, 132.6, 130.5, 130.3, 130.1, 125.0, 123.0, 118.7, 117.9, 115.6, 114.6, 79.2, 61.4, 60.3, 30.9, 30.7, 23.7, 23.5 ppm. ^195^Pt{^1^H} NMR (DMSO-d_6_, 86 MHz): *δ* = 1410 ppm. ^19^F{^1^H} NMR (DMSO-d_6_, 376 MHz): *δ* = −61.31 ppm. ESI-HRMS^+^: *m/z* [M+H]^+^ calc. for C_22_H_25_F_3_N_3_O_7_^194^Pt: 694.1266, found: 694.1288. EA: calc. for C_22_H_24_F_3_N_3_O_7_Pt · 1/7 CHCl_3_: C: 37.38%; H: 3.42%; N: 5.91%, found: C: 37.46%; H: 3.82%; N: 5.78%.

*Cis,cis,trans*-[Pt(DACH)(ox)(TOLF-H)OH] (**10**). Synthesis was performed according to general procedure 2. **O2** (150 mg, 0.35 mmol), **3** (145 mg, 0.38 mmol). Yield: 25 mg (10.6%) as yellow solid. ^1^H NMR (DMSO-d_6_, 400 MHz): *δ* = 9.44 (s, 1 H), 8.24 (m, 2 H), 7.96 (m, 1 H), 7.86 (dd, *J* = 7.9/1.4 Hz, 1 H), 7.23 (m, 5 H), 6.81 (d, *J* = 8.3 Hz, 1 H), 6.73 (t, *J* = 7.4 Hz, 1 H), 2.67 (m, 2 H), 2.24 (s, 3 H), 2.10 (m, 2 H), 1.48 (m, 4 H), 1.12 (m, 2 H) ppm. ^13^C{^1^H} NMR (DMSO-d_6_, 101 MHz): *δ* = 175.6, 163.9, 145.9, 141.2, 134.4, 132.9, 132.5, 129.3, 127.4, 124.1, 121.0, 117.3, 117.0, 113.7, 65.0, 60.3, 30.9, 30.7, 23.7, 23.6, 14.8 ppm. ^195^Pt{^1^H} NMR (DMSO-d_6_, 86 MHz): *δ* = 1414 ppm. ESI-HRMS^+^: *m/z* [M+H]^+^ calc. for C_22_H_27_ClN_3_O_7_^194^Pt: 674.1159, found: 674.1173. EA: calc. for C_22_H_26_ClN_3_O_7_Pt · 1/6 CHCl_3_: C: 38.31%; H: 3.80%; N: 6.05%, found: C: 38.34%; H: 3.96%; N: 5.97%.

*Cis,cis,trans*- [Pt(DACH)(ox)(FEL-H)OH] (**11**). Synthesis was performed according to general procedure 2. **O2** (100 mg, 0.23 mmol), **4** (79 mg, 0.26 mmol). Yield: 55 mg (38.2%) as white solid. ^1^H NMR (DMSO-d_6_, 400 MHz): *δ* = 8.32 (m, 1 H), 8.14 (m, 1 H), 7.81 (m, 1 H), 7.62 (d, *J* = 7.4 Hz, 2 H), 7.53 (d, *J* = 8.1 Hz, 2 H), 7.45 (t, *J* = 7.6 Hz, 2 H), 7.34 (t, *J* = 7.3 Hz, 1 H), 7.29 (d, *J* = 8.1 Hz, 2 H), 7.09 (m, 1 H), 3.57 (s, 2 H), 2.40 (m, 2 H), 2.05 (m, 2 H), 1.45 (m, 3 H), 1.26 (m, 1 H), 1.07 (m, 2 H) ppm. ^13^C{^1^H} NMR (DMSO-d_6_, 101 MHz): *δ* = 179.9, 164.0, 140.2, 138.3, 135.7, 129.9, 129.0, 127.4, 126.6, 126.5, 61.5, 60.1, 43.3, 30.9, 23.7 ppm. ^195^Pt{^1^H} NMR (DMSO-d_6_, 86 MHz): *δ* = 1409 ppm. ESI-HRMS^+^: *m*/*z* [M+H]^+^ calc. for C_22_H_26_N_2_O_7_^194^Pt: 625.1440, found: 625.1447. EA: calc. for C_22_H_26_N_2_O_7_Pt ∙ 1/9 CHCl_3_: C: 41.57%; H: 4.12%; N: 4.39%, found: C: 41.78%; H: 3.78%; N: 4.46%.

General procedure 3 for further modification of platinum(IV) complexes:

Complex **11** (1 equiv.) and the acid anhydride (3 equiv.) were stirred in DMF (5 mL) at room temperature for 3 days. The raw product was purified by column chromatography (chloroform/methanol).

*Cis,cis,trans*-[Pt(DACH)(ox)(FEL-H)(OAc)] (**12**). Synthesis was performed according to general procedure 3. **11** (110 mg, 0.18 mmol), acetic anhydride (54 mg, 0.53 mmol). Yield: 40 mg (33.3%) as white solid. ^1^H NMR (DMSO-d_6_, 400 MHz): *δ* = 8.20 (m, 4 H), 7.63 (d, *J* = 8.4 Hz, 2 H), 7.55 (d, *J* = 8.2 Hz, 2 H), 7.45 (t, *J* = 7.6 Hz, 2 H), 7.33 (dd, *J* = 17.8/7.8 Hz, 3 H), 3.64 (d, *J* = 3.6 Hz, 2 H), 2.55 (m, 1 H), 2.40 (m, 1 H), 2.06 (dd, *J* = 27.1/0.8 Hz, 2 H), 1.95 (s, 3 H), 1.35 (m, 4 H), 1.08 (m, 2 H) ppm. ^13^C{^1^H} NMR (DMSO-d_6_, 101 MHz): *δ* = 178.6, 178.2, 163.4, 163.4, 140.0, 138.3, 135.0, 129.9, 128.9, 127.3, 126.5, 126.4, 60.9, 60.8, 42.1, 31.0, 30.7, 23.5, 23.4, 23.0 ppm. ^195^Pt{^1^H} NMR (DMSO-d_6_, 86 MHz): *δ* = 1613 ppm. ESI-HRMS^+^: *m/z* [M+H]^+^ calc. for C_24_H_29_N_2_O_8_^194^Pt: 667.1545, found: 667.1540. EA: calc. for C_24_H_28_N_2_O_8_Pt: C: 43.18%; H: 4.23%; N: 4.20%, found: C: 42.91%; H: 4.29%; N: 4.18%.

*Cis,cis,trans*-[Pt(DACH)(ox)(FEL-H)(PhBu)] (**13**). Synthesis was performed according to general procedure 3. **11** (110 mg, 0.18 mmol), phenyl butyric anhydride (164 mg, 0.53 mmol). Yield: 50 mg (36.8%) as white solid. ^1^H NMR (DMSO-d_6_, 400 MHz): *δ* = 8.35 (m, 3 H), 8.07 (m, 1 H), 7.63 (m, 2 H), 7.55 (d, *J* = 8.2 Hz, 2 H), 7.45 (t, *J* = 7.6 Hz, 2 H), 7.34 (m, 3 H), 7.26 (t, *J* = 7.4 Hz, 2 H), 7.16 (t, *J* = 7.7 Hz, 4 H), 3.65 (d, *J* = 3.5 Hz, 2 H), 2.44 (m, 2 H), 2.26 (t, *J* = 7.6 Hz, 2 H), 2.07 (m, 2 H), 1.73 (m, 3 H), 1.40 (m, 4 H), 1.05 (m, 2 H) ppm. ^13^C{^1^H} NMR (DMSO-d_6_, 101 MHz): *δ* = 180.8, 178.2, 163.5, 163.5, 141.6, 140.0, 138.3, 135.1, 130.0, 129.0, 128.4, 128.3, 127.3, 126.6, 126.5, 125.8, 61.1, 60.8, 42.1, 35.2, 34.3, 31.0, 30.8, 27.2, 23.5 ppm. ^195^Pt{^1^H} NMR (DMSO-d_6_, 86 MHz): *δ* = 1607 ppm. ESI-HRMS^+^: *m*/*z* [M+H]^+^ calc. for C_32_H_37_N_2_O_8_^194^Pt: 771.2171, found: 771.2174. EA: calc. for C_32_H_36_N_2_O_8_Pt: C: 49.80%; H: 4.70%; N: 3.63%, found: C: 49.51%; H: 4.70%; N: 3.66%.

### 3.3. Stability Studies

Complex **7** was dissolved in 2% DMF in aqueous phosphate buffer (5 mM, pH 7.4) to a concentration of 0.02 mM and was reacted with a 0.2 mM solution of ascorbic acid in aqueous phosphate buffer in a 1:1 mixture. The reaction mixture was incubated at 37 °C while samples for UHPLC-HRMS were repeatedly analyzed over 12 h.

Next, 100 µM solutions (0.1% DMSO) of compound **7** in PBS, 12 and 120 mM NaCl, DMSO, 150 µM BSA, and 1 mg/mL salmon sperm DNA were monitored by UV-Vis spectroscopy.

### 3.4. Cell Culture Conditions

Ovarian carcinoma cell lines A2780 and SKOV3, as well as their cisplatin-resistant cell lines A2780cis and SKOV3cis were cultivated in RPMI 1640 medium, supplemented with 10% fetal calf serum, 100 U/mL penicillin, and 100 μg/mL streptomycin (Life Technologies, Darmstadt, Germany) in an incubator (37 °C, 5% CO_2_, 90% humidity). For biological experiments, cells were counted, seeded on different well plates, and incubated overnight at 37 °C to enable cell attachment. Platinum-resistant A2780 and SKOV3 cells were established as described [82] by repeated rounds of 3-day incubations with increasing amounts of cisplatin starting with 0.1 µM. The concentration was doubled after 3 incubations interrupted by recovery phases with normal medium. Cells that survived the third round of 12.8 µM cisplatin were defined as resistant cultures. Resistant cultures were not steadily exposed to cisplatin for maintaining the resistant phenotype avoiding the accumulation of additional (epi-)genetic changes caused by cisplatin. However, early cryo stocks were used and IC_50_ values of resistant cultures were stable over time.

### 3.5. Determination of IC_50_ Values

Cells were seeded on 96-well plates (5000 cells per well for SKOV3 cell lines, 10,000 cells per well for A2780 cell lines). The platinum(IV) complexes **5**–**8** and **10**–**13** and the free NSAIDs were dissolved in DMSO, serially diluted in DMSO, and afterwards diluted in RPMI 1640 medium to receive the concentrations from 0.1 to 100 μM. Cisplatin and oxaliplatin were dissolved in 0.9% NaCl solution right before the experiment and diluted serially in RPMI 1640 medium. The samples were added in 200 μL per well in triplicates and were incubated at 37 °C, 5% CO_2_ for 48 h. Viable cells were determined by MTT (Promega, Walldorf, Germany) assays (incubation time 2 h). After subtraction of blank MTT values, relative values were compared to the mean of the medium controls. Non-linear regression analyses using the Hill-slope were accomplished with GraphPad 5.0 software (Dotmatics, Boston, MA, USA).

### 3.6. Analyses with Flow-Cytometry

Cells were seeded on 6-well plates (150,000 cells per well). Complex **7** was dissolved in DMSO and diluted in RPMI 1640 medium to receive concentrations from 0.1 to 2 μM. The samples were added in 3 mL per well and incubated at 37 °C, 5% CO_2_ for 48 h. For cell death experiments: Adherent and suspended cells were collected and stained with 1 μg/mL propidium iodide (PI) solution in PBS directly before measurement. For cell cycle experiments: Cells were trypsinized, washed, and fixed in 70% ice-cold ethanol at −20 °C for at least 3 h. After two washing steps with PBS, cells were incubated for 45 min at 4 °C with RNase buffer and stained with 50 μg/mL PI directly before measurement. Flow cytometry measurements analyzed at least 10,000 single cell events.

### 3.7. Real-Time-Glo^TM^ Annexin V Apoptosis and Necrosis Assay (Promega, Catalog no. JA1011)

The assay was performed according to the manufacturer’s instructions (Promega, Walldorf, Germany). Cells were seeded on a 96-well plate (5000 cells per well for SKOV3 cell lines, 10,000 cells per well for A2780 cell lines). Complex **7** was dissolved in DMSO and diluted in RPMI 1640 medium to reach the stated concentrations, which were added in 100 μL. Untreated cells in RPMI 1640 medium served as control. Next, 100 μL of the 2X detection reagent (containing 12 mL RPMI 1640 medium, 24 μL Annexin NanoBiT^®^ substrate, 24 μL CaCl_2_, 24 μL necrosis detection reagent, 24 μL Annexin V-SmBiT and 24 μL Annexin V-LgBiT) were added per well. The plate was shaken for 30 s, and both luminescence and fluorescence measurements were conducted at various time points over 70 h with a Tecan plate reader M200pro. The excitation wavelength was set at 485 nm, the emission wavelength was set to 530 nm.

### 3.8. COX (Ovine/Human) Inhibitor Screening Assay (Cayman Chemical, Item no. 560131)

According to the instructions of the manufacturer (Cayman Chemical, Ann Arbor, MI, USA), the assay was composed of three main parts: COX reactions, standard preparation, and enzyme-linked immunosorbent assay (ELISA). Complex **7** (dissolved in DMSO) and cisplatin (dissolved in 0.9% NaCl solution) were examined for their ability to inhibit COX-1 and COX-2. Both compounds were diluted in RPMI 1640 medium to reach stated concentrations.

COX reactions: background samples (20 μL inactivated COX-1/2 enzyme), COX-1/2 100% initial activity samples, and COX inhibition samples (10 μL COX-1/2 enzyme, 10 μL COX inhibitor) were incubated with heme (10 μL) and reaction buffer (160 μL) for 10 min at 37 °C. The reaction with 10 μL arachidonic acid for 2 min at 37 °C was stopped with 30 μL stannous chloride. All samples were diluted with ELISA buffer (background: 1:100; COX-1/2 100% initial activity and COX inhibition: 1:2000, 1:4000).

Standard preparation: The prostaglandin screening standard was serially diluted in ELISA buffer.

ELISA: The format of the 96-well plate was adopted from the protocol: blank wells, non-specific binding wells (NSB, 100 μL ELISA buffer), maximum binding wells (B_0_, 50 μL ELISA buffer), prostaglandin screening standard (50 μL of each concentration in duplicate), background samples (50 μL in duplicate), COX-1/2 100% initial activity samples (50 μL in duplicate), COX-1/2 inhibitor samples (50 μL in duplicate), and an empty total activity well. Then, 50 μL prostaglandin screening acetylcholinesterase (AChE) tracer was added to all wells except for the blank and total activity wells. Next, 50 μL prostaglandin screening ELISA antiserum was added to each well except for the blank, NSB, and total activity wells. The plate, covered with plastic film, was incubated for 18 h at room temperature on an orbital shaker. The plate was emptied, and each well was rinsed with wash buffer five times. Next, 200 μL of Ellman’s reagent in ultrapure water was added to each well, and the AChE tracer (5 μL) was added to the total activity well. The plate, covered with plastic film, was developed for 30 min in the dark at room temperature on an orbital shaker. The absorbance was measured at 410 nm with a Tecan plate reader.

### 3.9. Quantitative PCR

According to the manufacturer’s instructions, total ribonucleic acid (RNA) was extracted from cells using NucleoSpin RNA Kit (MachereyNagel, Düren, Germany) including DNase treatment. RNA concentration was determined spectrophotometrically with NanoDrop ND-1000 (Thermo Fisher Scientific, Darmstadt, Germany). RNA (1 µg) was heat denatured (10 min, 70 °C) with an anchored poly-dT primer (1 pmol/μL; 5′-20xT-VN-3′) and dNTP (0.5 mM each) to enable annealing before adding, first strand buffer, DTT (0.1 M), RNaseOUT^TM^ (40 units), and SuperScript II reverse transcriptase (200 units, Invitrogen, Thermo Fisher Scientific, Darmstadt, Germany). The reaction was incubated for 60 min at 42 °C and stopped for 15 min at 70 °C.

The real-time PCR experiments were run on a Rotorgene cycler (Qiagen, Hilden, Germany). Reactions were performed using the FastStartUniversal SybrGreen Mastermix (Roche Diagnostics, Mannheim, Germany) containing forward and reverse primers (10 pmol each) and cDNA equivalent to 25 ng RNA. Primer-specific data are listed in Table 2. The PCR steps were as follows: initial denaturation and hot start activation at 98 °C for 10 min followed by 40 cycles of denaturation phase at 98 °C for 15 s, primer-specific annealing for 20 s at Ta, and elongation at 72 °C for 40 s. Subsequently, the melting temperature of the PCR product was determined to ensure specificity. Relative target gene expression was normalized to the expression of two housekeeping genes (*GAPDH*, *HPRT*) and calculated relative to untreated controls.

## 4. Conclusions

This work describes the synthesis and characterization of nine novel cisplatin- and oxaliplatin-based platinum(IV) complexes with different axial NSAID ligands. Whereas the free NSAIDs do not show relevant anticancer activity in the tested ovarian carcinoma cell lines, the corresponding platinum(IV) complexes containing the NSAIDs have significantly increased cytotoxicity. Specifically, the cisplatin-based complexes (**5**–**7**) exhibited a high cytotoxicity against both sensitive and resistant cell lines. Compared to the mono-substituted platinum(IV) complexes based on oxaliplatin, the triple-action complex **13** shows improved anticancer activity in all cell lines.

Due to the highest measured cytotoxicity, complex **7** was selected for further biological experiments. These experiments suggest that compound **7** acts independently of cisplatin-resistance mechanisms and causes similar molecular effects in cisplatin-sensitive and -resistant cell lines. Compound **7** induces a strong cytostatic effect and cell-line-dependent early apoptotic or late necrotic cell death processes in SKOV3 and A2780, respectively. Gene expression changes under compound **7** treatment agree with these effects and may point to the contribution of ATF3-mediated stress response to the biologic activity.

## Data Availability

The presented data are available in the article and in the Appendix A.

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
