# Peer review of "Synthesis, Characterization and Biological Investigation of the Platinum(IV) Tolfenamato Prodrug–Resolving Cisplatin-Resistance in Ovarian Carcinoma Cell Lines"

_ijms, 2023, doi:10.3390/ijms24065718_

Round 1

Reviewer 1 Report

The authors describe the synthesis of (nine) platinum(IV) complexes using a cisplatin or carboplatin core with four different NSAID’s to obtain cytotoxic compounds aiming to overcome cisplatin resistance in two ovarian cancer cell lines. Eight compounds were investigated with respect to cytotoxicity and one compound was selected for further cell biological assays. This is an interesting and potentially relevant approach. However, this study contributes to an intensively elaborated research field of similar focus and thus, lacks a novelty since the authors cannot really contribute further knowledge to understand the complex activity in resistance. In these terms, concerning the design of the study and the selection of the indicated NSAIDs, the authors do not explain why using felbinac, while the selection of the fenamates might be supported by the findings in Ref. 60 - 62.

Several major aspects have to be considered:

The cell biological experiments show that compound 7 possess cell toxicity in the indicated four cell lines with minor differences between sensitive and resistance cells. However, what makes comparison upon the assays and partly upon compound 7 and cisplatin difficult, if even possible is the difference in concentrations used. While apoptosis was checked using 0.1/0.2 µM and 1 or 2 µM of compound 7, cell cycle control used lower amounts and qPCR even other, higher concentrations. Why this? The concentrations are partly above the IC50 values in A2780 cells.

Concerning the qPCR data in Figure 4, no significances were show to validate any interpretations. Generally, cisplatin and compounds 7 seem not to show any differences in SKOV3 and SKOV3cis cells, valuable to interpret any differences to explain overcoming resistance. In A2780cells, the interpretation of p21 expression (line 275…, p. 8) showing lower expression upon cisplatin; have other concentrations been checked? In this case, while 7 is again above the IC50 value, cisplatin in at roughly half of IC50 in A2780 and only by 20% of A2780cis cells.  This might be a reason for the lower cisplatin activity. The authors do not refer to these fact.

Figure 1, the legend is insufficient and the abbreviations were not explained.

Suggestion for further experiments:

Since the redox status of the cells seems to be a key aspect in resistance, have the authors investigated ROS upon treatment?, or GSH to probable get insight into mechanisms how to break resistance. Those data appear essential for interpretation.

Have the authors checked the intracelular platinum concentrations to compare them with activity (restrictions).

Minor points:

The Introduction is partly sprawling with facts concidered to be superfluous in this type of study, references (>70 for Intro) are even exaggerated to explain established findings. In contrast, the discussion of the findings is even weak.

Author Response

Dear Reviewer 1,

Best wishes

Wolfgang Weigand

Reviewer 2 Report

Reviewer Summary:

In this manuscript, Barth et al., describe the synthesis of a set of platinum(IV) complexes that integrate NSAID axial ligands and their biological evaluation in ovarian cancer cell lines with cisplatin-sensitive and -resistant characteristics.  Although the inclusion of NSAIDs as axial ligands in Pt(IV) compounds is not partucularly novel, the complexes described in the project have merit and are of interest in the field.  A strength of the manuscript is the description of the chemical synthesis and validation.  Points of weakness in the manuscript include: several of the biological assays are not sufficiently explained, there are frequent grammatical and clarity issues present in the text, and most of the chemical structure drawings are not of acceptable quality.  I have provided below a list of comments, that if addressed, would be sufficient for this reviewer to authorize publication of the manuscript.

Comments:

[lines 18-19] change to: Platinum(IV) fenamato complexes with a cisplatin-core showed especially high in vitro cytotoxicity against the tested cell lines.

[line 19] change “The most promising complex 7 was further” to “The most promising complex, 7, was further”

[line 21] change “and,” to “, and”

[line 22-24] change “Gene expression changes under…..to the biologic activity” to “Gene expression analysis suggests that compound 7 acts through a stress response pathway integrating p21, CHOP, and ATF3.

[lines 38-41] change “Despite their abundant…..other nucleophiles [15,16,19-23]” to “Despite their abundant clinical benefit, cisplatin and oxaliplatin treatment can also cause resistance and severe side-effects [15,16,19-23].”

[lines 53-55] remove “researchers’” and “fast” from the sentence.

[line 72] change “implicates” to “are the”

[line 73] change “have already shown to” to “can”

[lines 80-82] change “However, the conjugation of FEL ….. unexploited potential.” to “However, the conjugation of FEL and the fenamates to a platinum(IV) complex and their effect on cytotoxic activity has not yet been investigated.”

[lines 91-95] change “The used model systems of isogenic ovarian ….. low survival rate below 40 %. “ to “ “The ovarian cancer cell lines used in this study allow analysis of resistance mechanisms and are of clinical relevance because the majority of ovarian cancer patients eventually develop resistance to platinum compounds contributing to a low survival rate below 40 %.”  Also, there should be a reference to ovarian cancer resistance to platinum treatment (cisplatin and oxaliplatin) and to the 40% survival rate statistic.

[line 112, Scheme 1] The quality of the chemical structure drawings shown in Scheme 1 is poor.  For example, many of the lines showing bonds are dashed in appearance making them barely legible.  All chemical bonds, except for cis bonds, should be drawn using solid, legible lines (Figure S28 in the Supplementary Materials provides examples of acceptable high quality structures using solid, legible lines).  Some bonds are also missing in Scheme 1, e.g., in A., the left-hand most ring structure, in A., compound 1, the lowest ring structure isn’t closed, and in A., compound 3, the lower bond in the amine bridge is missing making it look like two separate compounds [In summary, compounds 1, 3, 8, 11, 12, and 13 have missing bonds]. Also, there appears to be a partial arrow head that was cut-off immediately above the F3C substituent in compound 6, and arrows should be made bolder and more legible throughout.  More detail of the synthesis of 11 should be provided as some of the intermediates don’t seem to be provided.  The axial ligands that were added to 11 to make 12 and 13 should be shown separately from the final products to make it easier for the reader to understand this step of the synthesis.

[lines 151-152] The expression: “The safety of cells was preserved by maintaining the DMSO content of all concentrations at 0.1 %.” is unclear.  Rephrase this sentence to say, “To prevent damage to cells, the concentration of DMSO in cell culture experiments was limited to 0.1%”.

[lines 157-159] Change “Whereas the free NSAIDs ….and the axial ligands.” to “Whereas the free NSAIDs do not cause appreciable cytotoxicity (IC50 > 100 μM; Table 1), when attached to a platinum(II) core, the resulting platinum(IV) complexes cause much lower IC50 values in the cancer cell lines tested (Table 1).”

[lines 157-176] At the end of every sentence where an IC50 value is being referenced, insert “(Table 1)”.

[line 165] When “RF” first appears in the text, it should be defined as “resistance factor”.

[line 178-179] Define resistance factor mathematically as “IC50 value resistant cell line/IC50 value parental cell line”.

[line 175] Was there a reason why compound 5 was not selected? It has very similar IC50 values compared to compound 7. Please state the reason if there is one.

[line 180] The statement: “IC50 values substantially higher than 100 µM, just determined computationally by exploration” is very underdeveloped.  The authors should state here (or in the methods section) precisely how the computational analysis was done by providing the identity of the software that was used with manufacturer source information (city, country) and what specific computational procedure was used.

[line 182] change “To figure out more details about the stability behavior….” to “To analyze the stability behavior…”

[line 199] Please place a supporting reference at the end of the sentence.

[lines 202-203] The sentence: “Thus, the ligand TOLF must be released after reduction to platinum(II) to inhibit COX.” needs to be clarified.  In the prior sentence, the authors state that the “reduced complex #7 inhibited COX-1 activity significantly by ….. 47.4 %”. However, review of Supplemental Figure S29 shows that the 47.4% effect was found with complex #7 with 800 µM ascorbic acid, which is the reduction experiment described in sections 2.4. and 3.3., and shown in Supplemental Figure S28. This result does not necessarily prove that the ligand TOLF was released from the complex under the chemical conditions present within the COX activity assay reaction buffer, which is chemically different than the buffer solution used in the stability study in S28.  Instead, the ascorbic acid could be responsible for the effect on COX activity alone or in synergy with the 80 µM of complex 7.  Also, the absence of any measurable effect on COX-1 activity from complex 7 alone shown in S29 could be due to the complex failing to undergo reduction in this assay and, therefore, not releasing functional cisplatin and TOLF (a UHPLC-HRMS analysis of COX assay samples as in S28 could resolve this point and provide evidence whether a kit reaction mixture component is interfering with complex 7 reduction). Please rewrite this section to address these interpretations of the data and to provide a better discussion of what could cause the suboptimality of this assay.

[line 214] change to: “Cell death and cell cycle distribution analysis”

[line 219] “rate” is usually defined as a quantity per unit of something else, and Figure 1A presents the rate as a % value on the y-axis, which isn’t a rate.  Please either convert the y-axis into a rate value or change “rate” in the text to a percentage measure.

[line 223] change “death” to “dead”.

[line 225] define “2*IC50”.

[line 233] place a 0 next to Ap, Ac, Sp, and Sc on the x-axis for the untreated control samples.

[lines 234-236] define Ap, Ac, Sp, and Sc in the figure legend.

[line 248] change “on” to “of”

[lines 249-250] change “To analyze induced cell death …see Experimental part).” to “To analyze cell death, apoptosis and necrosis was measured in control and compound 7 treated cell lines using a commercial kit (see 3.7.).”.  Also, place information about the controls used in this experiment in 3.7.

[line 263] one trend in the data is that apoptosis decreases at later time points.  The authors should explain in the discussion why they believe this effect is observed.  For example, does the data suggest that resistance to complex 7 is beginning to occur later in the assay or that the number of live cells in the samples has decreased and is responsible for the attenuated signal (was this observed in the 96-well plates)?

[lines 264-266] change “Exemplary real-time measurement …. complex 7.” to “Real-time measurement of apoptosis and necrosis.” The following sentence, “Main differences were observed …not between parental and resistant cells.”, should be deleted as it is discussion and not a description of the figure.  Instead, information on statistical analysis and sample size should be placed here.

[lines 268-269] please provide a couple of supporting references to platinum compound PCR gene studies at the end of the sentence.

[line 291] replace “contrary” with “However”

[lines 301-536] in the Materials and Methods section, complete vendor information should be provided throughout. 

[line 305] change the second “of” to “to”

[line 361] remove the second “of”

[lines 462-468] Did the authors induce cisplatin resistance in the A2780cis and SKOV3cis cell lines?  This information needs to be provided here.  Typically, 1 µM cisplatin is added to the media every 2-3 passages to maintain resistance to cisplatin in the A2780 cell line, e.g., Almotairy et al., Pt(IV) pro-drugs with an axial HDAC inhibitor demonstrate……, J Inorg Biochem. 2020 Sep;210:111125, and 3 µM in the SKOV3cis cell line, e.g., Howard et al., Dinaciclib as an effective pan-cyclin dependent kinase inhibitor in platinum resistant ovarian cancer, Front. Oncol., 25 November 2022.  If this was not done, then the authors need to explain how the results in Table 1 are still valid or repeat the experiments to determine the proper relative IC50 ratios (the RF of 1.5 for CisPt in the SKOV3 cell line is rather low and not particularly convincing. See Huang et al., Role of Wnt/β-catenin, Wnt/c-Jun N-terminal kinase and Wnt/Ca2+ pathways in cisplatin-induced chemoresistance in ovarian cancer, Exp Ther Med. 2016 Dec;12(6):3851-3858., where they report an IC50 RF of 4.3 for the SKOV3par and SKOV3cis (aka SKOV3/DDP) cell lines at 48 hours).

[lines 485-486] In this experiment it states: “Dead cells in the supernatant and trypsinized cells were combined and stained with 1 μg/mL propidium iodide (PI) solution”.  The cells suspended in the supernatant most likely represent a heterogenous population of live and dead cells.  Change the sentence to say, “Adherent and suspended cells were collected and stained with 1 μg/mL propidium iodide (PI) solution.” 

[lines 498-501] To simply say that “The instructions of the manufacturer were carried out.” is insufficient.  The authors need to describe here in detail what was done (how many cells were used per well, what kind of plate, how long the experiment was run, etc.).    

[lines 528-532] change “Thus, and due to its high activity ……between parental and cisplatin-resistant cells.” to “Due to its high cytotoxicity, complex 7 was selected for further biological experiments. These experiments suggest that compound 7 acts independently of cisplatin resistance mechanisms and causes similar molecular effects in cisplatin-parental and -resistant cell lines”.

The Supplementary Materials are excellent with the exception of the chemical structure diagrams in S1-S26.  These need to be replaced with higher quality diagrams like those shown in the preparation of O1 and O2, and in S28, that legibly show all of the bonds and substituent groups.

Author Response

Dear Reviewer 2,

Best wishes

Wolfgang Weigand

Reviewer 3 Report

Marie-Christin Barth et al. proposed Synthesis, characterization and biological investigation of the platinum(IV) tolfenamato prodrug – resolving cisplatin-resistance in ovarian carcinoma cell lines.

The studies with experimental findings are well planned and executed by the authors. I recommend that this article be published in its current form.

Author Response

Dear Reviewer 3,

Best wishes

Wolfgang Weigand

Reviewer 4 Report

In this manuscript, the authors gave an attractive story about design, synthesis, and biological activity of cisplatin- and oxaliplatin-based platinum(IV) complexes with four different NSAID.  I would recommend this manuscript to be published. However, there are some issues that require improvement or clarification.

1.       Under the introduction, a figure for platinum compounds and NSIDS should be included.

2.       The aim of the study should be represented in a figure.

3.       The resolution of scheme on should be improved.

4.       Line 145 should be written under the experimental section.

5.       Cytotoxicity study should be done on at least one normal cell line.

6.       The title of 2.5. needs to be rewritten as In vitro inhibition of COX activity.

7.       The title of 2.6. needs to be rewritten as Cell cycle analysis and cell cycle distribution.

8.       The title of 2.8. also needs to be rewritten.

9.       The conclusion is not well written. It should be improved.

Author Response

Dear Reviewer 4,

Best wishes

Wolfgang Weigand

Reviewer 5 Report

1.      Abstract. The introductory part was complete ignored.

2.      The methods shall be clearly explain, the results as well, the results shall be quantitative. Then draw the conclusion in connection with the results

3.      Line 16-17, rewrite the sentence

4.      Usually it is preferred to write in the passive form, For example, in line No. 82 "we show", and follow throughout the manuscript

5.      Line 43-49, these paragraphs shall be before the objective of the study to provide the strong rationale to present research

6.      The results and discussions section needs to be improved to include the discussion on the results obtained.

Author Response

Dear Reviewer 5,

Best wishes

Wolfgang Weigand

Round 2

Reviewer 1 Report

The authors have improved the paper and replied to the concerns of the reviewer. Concerning the qPCR data in Figure 4, although the authors added some further data using higher cisplatin concentrations, these data do not really allow a general conclusions for a better activity of 7 compared to cisplatin, unless a concentration dependency or another mode of action. This is also restricted by a missing statistical evaluation, which is not even possible at n = 2. It would be better, if not requesting more experiments, to at least critically refer to this in the text.

Remaining typos in the corrected version, e.g. line 177 should be considered.

Author Response

Dear Reviewer

Thank you for the handling and fast processing of our manuscript. Please find attached our revised version R2 after minor revision. We agree with the reviewer that we should add the missing statistical validation as limitation of the real-time PCR experiments and added following statement to the manuscript (page 9, line 603): “However, additional experiments are necessary to (i) enable a statistical evaluation, which is not possible with the presently available data of n=2 independent treatments and (ii) to determine the exact mechanism for the activation of the ER stress pathway by compound 7.” In addition, we checked the text for typos and corrected them.

Best regards

Wolfgang Weigand

Reviewer 4 Report

Authors have done all suggestions and answer all concerns. I suggest publishing in the present form.

Author Response

We thank the reviewer for the kind message.

Wolfgang Weigand